# Unanchoring the Mind: DAE-Guided Counterfactual Reasoning for Rare Disease Diagnosis

## Abstract

Diagnosing rare diseases remains a persistent challenge, often hindered by *cognitive anchoring*: once clinicians settle on a common diagnosis, alternative-especially rare-explanations are often dismissed. To address this, we propose a human-centered counterfactual reasoning framework using a Denoising Autoencoder (DAE) to simulate *what-if* diagnostic scenarios that disrupt clinicians' initial assumptions. Our model uniquely jointly learns (1) the true distribution of diseases and symptoms, and (2) human diagnostic behavior, revealing critical gaps between *medically possible* and *clinically considered* diagnoses. By strategically perturbing latent patient representations, it generates *contrastive counterfactuals* that highlight rare-but-plausible conditions—conditions typically overlooked due to cognitive bias. Unlike traditional decision-support tools, our system *proactively* suggests rare diseases not because they are statistically probable, but because they are *cognitively neglected*. Evaluated on three rare disease datasets, our approach outperforms standard machine learning classifiers in detecting rare conditions while maintaining strong performance on common diagnoses. Beyond boosting accuracy, it fosters *hypothesis-driven reasoning*, enhancing both diagnostic precision and clinician learning.

## 1 Introduction

Despite advances in machine learning for clinical diagnosis, *rare diseases remain notoriously difficult to identify* due to their low prevalence, heterogeneous manifestations, and frequent overlap with more common conditions (Schieppati et al., 2008; Griggs et al., 2009). Consider a patient presenting with persistent fatigue, joint pain, and skin rashes, clinicians often anchor on familiar diagnoses like lupus rather than considering rare alternatives such as Ehlers-Danlos syndrome. This diagnostic misdirection is not merely a result of statistical rarity or symptom ambiguity, but also due to a well-documented *cognitive bias* known as *anchoring*—clinicians' tendency to settle prematurely on an initial diagnosis and insufficiently revise it in light of new or contradictory evidence (Tversky & Kahneman, 1974; Saposnik et al., 2016; Croskerry, 2002; Li et al., 2023).

This *cognitive anchoring* introduces a significant bottleneck in *rare disease detection*, often leading to prolonged diagnostic delays, repeated misdiagnoses, and unnecessary interventions. Studies in clinical cognition have shown that medical decision-making is often driven by fast, heuristic-based thinking that prioritizes pattern recognition over analytical reassessment (Norman et al., 2024). This is especially problematic in the context of rare diseases, where diagnostic presentations often overlap with more common syndromes, creating fertile ground for premature closure. While previous machine learning efforts have primarily focused on enhancing accuracy through larger datasets or more powerful models (Juba & Le, 2019; Sun et al., 2017; Moreno-Barea et al., 2020), few have addressed the cognitive constraints that shape clinicians' interactions with model predictions, particularly under uncertainty. Moreover, existing studies indicate that clinicians may be unable to effectively integrate the AI's reasoning due to its opaque recommendations (Jussupow et al., 2021; Lebovitz et al., 2022), potentially exacerbating misdiagnoses (Jussupow et al., 2022).

Our work tackles the dual challenge of data sparsity and cognitive rigidity by introducing a diagnostic framework that not only *detects rare diseases* but also *mitigates the cognitive biases*—particularly *anchoring*—that hinder accurate diagnosis. Instead of merely maximizing predictive likelihood, our system acts as a cognitive aid, encouraging clinicians to consider alternative diagnostic hypotheses.

Drawing from cognitive science theories of bias mitigation (Croskerry, 2002) and leveraging recent advances in generative modeling, we design a Denoising Autoencoder (DAE) (Vincent et al., 2008) generative model to generate plausible diagnostic counterfactuals that promote reflective reasoning.

Our DAE-based model is trained on annotated clinical data to learn both disease distributions and typical diagnostic behaviors. By perturbing the latent representation of a patient's profile, the model generates alternative diagnostic paths—plausible yet cognitively overlooked possibilities—that suggest *follow-up tests*, outside the clinician's immediate expectations. For example, it might suggest:

> *The most likely rare disease overlapping with the current symptoms is **Ehlers-Danlos syndrome**. Consider additional tests such as **genetic screening** for connective tissue disorders. If the results are **positive**, the probability of this diagnosis **increases significantly***.

Unlike traditional AI systems that deliver static predictions, our framework promotes active cognitive engagement, helping clinicians *break habitual diagnostic patterns* and *rethink their assumptions*. By surfacing rare yet plausible conditions, it expands the diagnostic space, fosters reflective thinking, and supports more informed clinical decisions. As (Buçinca et al., 2021) have demonstrated, a mechanism that guides users to actively engage in critical thinking about initial assumptions enhances decision-making quality more effectively than merely providing predictions.

In our experiments, we evaluate the system's effectiveness using three rare disease datasets. our method outperformed conventional machine learning (ML) classifiers in rare disease detection while preserving optimal performance on common disease diagnosis. Counterfactual *validation* was performed by comparing the model's hypotheses with diagnoses made by *human clinicians* and assessments from *Large Language Models (LLMs)*. The results confirmed that our model could identify plausible but cognitively neglected conditions, thereby enhancing diagnostic precision and fostering clinician learning.

## 2 INHERENT CHALLENGES IN MODELING RARE DISEASE DIAGNOSIS

In clinical diagnosis, the fundamental task is to infer the underlying disease label $Y \in \mathcal{Y}$ from observed clinical evidence $X \in \mathcal{X}$, such as patient-reported symptoms. Both human clinicians and ML models aim to learn or approximate the mapping:

$$h : X \mapsto \hat{Y}, \quad \text{where } \hat{Y} \approx \arg\max_{Y} P(Y \mid X).$$

By Bayes' theorem, this conditional probability can be expressed as:

$$P(Y \mid X) = \frac{P(X \mid Y) \cdot P(Y)}{P(X)},$$

where $P(Y)$ encodes prior knowledge of disease prevalence and $P(X \mid Y)$ reflects the data-generating process (e.g., symptom presentation) conditioned on a specific disease. However, in the context of *rare disease diagnosis*, this inferential process becomes fundamentally challenging, no matter for logistic regression, support vector machines, or even deep classifiers, are all subject to the same three critical limitations:

1. **Skewed priors.** Rare diseases typically have extremely small $P(Y)$. This prior imbalance biases both clinicians and ML models to favor common diagnoses, even when rare diseases are more plausible explanations.

2. **Overlapping symptom profiles.** Many hallmark symptoms of rare diseases (e.g., fatigue, muscle pain, or inflammation) are nonspecific and widely shared across common conditions. As a result, the likelihoods $P(X \mid Y_{\text{rare}})$ and $P(X \mid Y_{\text{common}})$ often overlap significantly, making discrimination between them highly uncertain.

3. **Incomplete evidence.** Key diagnostic features—such as genetic markers or specialized imaging—are frequently missing from the record, due to cost, lack of access, or simply being overlooked. This leads to an underspecified $X$, causing both humans and machines to rely on incomplete or biased feature sets. Such gaps often *reinforce* cognitive heuristics like *anchoring*, where initial impressions dominate the diagnostic path.

These challenges create a shared *algorithmic–cognitive bottleneck* across both humans and machines. Standard discriminative models $h : X \mapsto Y$, trained to directly map observed features to labels, inherit the same structural vulnerabilities as their human counterparts. Without mechanisms to uncover latent structures, handle missing information, or actively de-bias the inference process, both fall short in the critical task of detecting rare and underrepresented diseases.

## 2.1 Motivation for a Latent-State Generative Model

These insights motivate the need for a new kind of AI-aided diagnostic framework—one that can:

- *Explicitly identify cases where the observed $X$ lies in an ambiguous or overlapping region* of the feature space;
- *Hypothesize possible latent rare disease explanations* even when current evidence is incomplete;
- *Proactively recommend additional complementary tests* (e.g., genetic panels, imaging) that can disambiguate competing diagnoses and help clinicians *break out of anchored diagnostic pathways*.

A discriminative model alone cannot meet these goals, as it is designed only to map observed input $X$ to a label prediction $\hat{Y}$ and lacks any mechanism for reasoning about uncertainty, missing data, or counterfactual information acquisition. To address these limitations, we propose a *latent-state generative model* based on the Denoising Autoencoder (DAE) framework. This model explicitly learns a latent representation $Z$ of the patient's symptom input $X$ and generates possible reconstructions and diagnostic outcomes in a controlled, interpretable manner. The goal is to assist both machine and human diagnostic reasoning by generating alternative hypotheses—especially those corresponding to rare conditions that might be missed due to low priors or heuristic bias.

The proposed latent-state generative model takes the following form (as illustrated in Figure.1):

- **Input:** $X$ (observed patient symptoms)
- **Latent state:** $Z$ (learned stochastic representation of patient condition)
- **Outputs:**
    1. $X'$: A reconstructed or generated version of patient symptoms (counterfactual or prototypical symptom set)
    2. $\hat{Y}^{\text{AI}}$: Prediction of the true diagnosis based on latent state $Z$
    3. $\hat{Y}^{\text{human}}$: Model's simulation of a human doctor's likely diagnostic decision

## 3 Our Proposed Generative Model Formulation

We assume access to a dataset of triplets $\left\{ \left( X_i, Y_i^{\text{human}}, Y_i^{\text{true}} \right) \right\}_{i=1}^{N}$, where $X_i \in \mathbb{R}^d$ represents patient features, $Y_i^{\text{true}} \in \{1, \ldots, C\}$ is the ground-truth diagnosis, and $Y_i^{\text{human}}$ is the clinician's recorded label. Our goal is to learn a generative latent-state model that captures three components: the patient's latent diagnostic state $Z$, the clinician's decision $Y^{\text{human}}$, and the AI's prediction $Y^{\text{AI}}$. By explicitly modeling the cognitive gap between human and AI reasoning, the model enables discrepancy-aware inference and supports bias-aware diagnostic support.

$$p_\theta \left( X, Y^{\text{AI}}, Y^{\text{human}}, Z \right) = p(Z) p_\theta(X \mid Z) p_\theta \left( Y^{\text{AI}} \mid Z \right) p_\theta \left( Y^{\text{human}} \mid \tilde{Z} \right) \tag{1}$$

Here, $Z \in \mathbb{R}^k$ is a latent representation inferred from $X$, and $\tilde{Z}$ denotes a modulated version of $Z$. Although humans and AI observe the same input $X$, their predictions can diverge due to: (1) cognitive load limiting human attention to parts of $X$, and (2) fundamentally different mapping functions. We explicitly reflect these factors in the design of our DAE-based generative model.

**Latent Representation Learning with Masked Denoising Autoencoder** Given that real-world clinical inputs $X \in \mathbb{R}^d$ often contain missing or underreported features, particularly for rare diseases, we employ a masked Denoising Autoencoder (mDAE) (Dupuy et al., 2024) strategy, to learn a robust and informative latent representation $Z$.

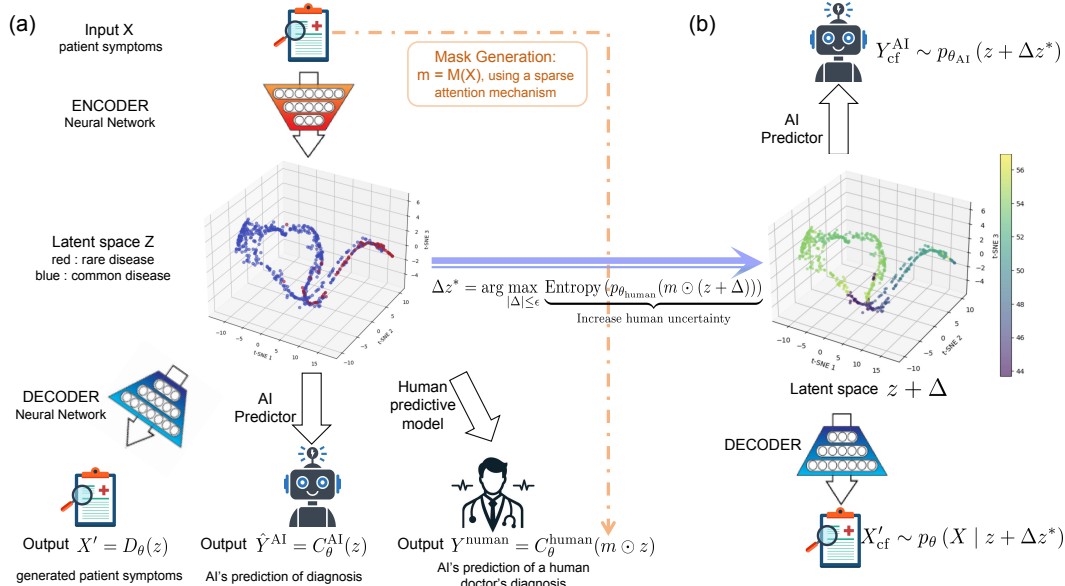

Figure 1: DAE-guided counterfactual reasoning framework. (a) DAE encodes patient features into a latent space, enabling dual predictors for AI and clinician diagnoses. (b) Counterfactuals are generated by perturbing latent vectors to increase uncertainty in human model. Then AI predictor can generate counterfactual diagnosis and decoder reconstructs the corresponding patient profile.

For each observed input $X_i$, we sample a binary mask $r_i \in \{0,1\}^d$ to randomly drop a subset of observed entries, simulating incomplete or noisy records. The resulting corrupted input is $\tilde{X}_i = r_i \odot X_i$, which is then encoded to a latent distribution $q_\phi\left(Z_i \mid \tilde{X}_i\right)$. The decoder reconstructs the full input, and the reconstruction loss is computed only on the originally observed (i.e., uncorrupted) entries:

$$\mathcal{L}_{\text{recon}} = \mathbb{E}_{q_\phi\left(Z_i \mid \tilde{X}_i\right)}\left[\left\|(1 - r_i) \odot \left(X_i - \hat{X}_i\right)\right\|_2^2\right] \tag{2}$$

This approach helps the model infer missing or overlooked features—like masked token prediction in language models—while learning robust, task-relevant representations. These generalizable embeddings enable effective downstream applications such as diagnosis prediction and modeling human-AI divergence.

**Dual Classification Losses**  The latent code $Z_i$ is leveraged to predict two diagnostic outcomes: the *ground-truth diagnosis* $Y_i^{\text{true}}$, and the *observed human diagnosis* $Y_i^{\text{human}}$.
We define two separate classification objectives:

- **AI Prediction Loss (truth-matching):**

$$\mathcal{L}_{\text{AI}} = -\mathbb{E}_{q_\phi(Z_i \mid X_i)}\left[\sum_c \alpha_c (1 - p_c)^\gamma \log p_c\right], \quad \alpha_c \propto \frac{1}{\text{freq}(c)} \tag{3}$$

Here, $p_c = p_{\theta_{\text{AI}}}\left(Y_i^{\text{true}} = c \mid Z_i\right)$ denotes the predicted probability of class $c$ under the AI classifier. This objective encourages the model to leverage the *full latent representation $Z_i$* to generate accurate, clinically grounded predictions aligned with the ground-truth diagnosis, using a classifier parameterized by $\theta_{\text{AI}}$.

To address class imbalance-particularly prevalent in rare disease settings, we employ a focal loss variant (Lin et al., 2017) that dynamically down-weights well-represented, easily classified categories and emphasizes learning from rare or ambiguous cases. As the system is intended to assist clinicians

in complex diagnostic scenarios, this calibrated formulation promotes more *exploratory* AI behavior, enabling the model to surface atypical or underrecognized patterns that may otherwise be overlooked. Thus, the AI acts not only as a predictor but also as a discovery aid, supporting more comprehensive and inclusive clinical decision-making.

- **Human Simulation Loss (cognitive-matching):**

$$\mathcal{L}_{\text{human}} = \mathbb{E}_{q_\phi(Z_i|X_i)} \left[ -\log p_{\theta_{\text{human}}} \left( Y_i^{\text{human}} \mid \tilde{Z}_i \right) \right] \tag{4}$$

Here, $\tilde{Z}_i = m_i \odot Z_i$ is a selectively masked version of the latent vector, where the learned attention mask $m_i \in [0,1]^k$ gates which latent dimensions are used by the human prediction head. This reflects the idea that, given the same input $X_i$, *humans and AI may focus on different parts of the data and apply distinct cognitive functions to reach a diagnosis*.

Importantly, the prediction functions for AI and human simulation are parameterized separately, using $\theta_{\text{AI}}$ and $\theta_{\text{human}}$ respectively. This architectural asymmetry captures both attentional differences (via $m_i$) and functional differences in diagnostic reasoning, allowing us to explicitly model and analyze human-AI cognitive divergence.

**Modeling Human-AI Cognitive Gaps via Sparse Self-Attention Mask**  Specifically, we compute the attention mask $m_i$ using a learnable self-attention module:

$$m_i = \text{Softmax}\left( \frac{Q\left(X_i\right) K\left(X_i\right)^\top}{\sqrt{d}} \right) V\left(X_i\right) \tag{5}$$

where $Q(\cdot), K(\cdot), V(\cdot)$ are linear projections (as proposed in (Vaswani et al., 2017)) that produce query, key, and value vectors from the input $X_i$, and the output is pooled to form a $k$-dimensional attention vector. This attention mechanism identifies which latent features humans are likely to focus on, given the current case.

To ensure interpretability and mimic human cognitive constraints, we impose an $\ell_1$ sparsity penalty on the attention mask:

$$\mathcal{L}_{\text{mask}} = \lambda_{\text{mask}} \cdot \|m_i\|_1 \tag{6}$$

This encourages the human prediction head to rely on a small subset of salient features, reflecting *limited cognitive bandwidth* and enhancing the *interpretability* of human diagnostic pathways.

**Contrastive Learning for Rare Disease Separability**  To prevent rare disease embeddings from collapsing into common clusters, we introduce a contrastive loss:

$$\mathcal{L}_{\text{contrast}} = \sum_{(i,j,k)} \max\left(0,\ \delta + d(Z_i, Z_j) - d(Z_i, Z_k)\right),$$

where $Z_i$ and $Z_j$ are latent representations from the same rare disease class, and $Z_k$ is from a common disease class.

This loss encourages embeddings of the same rare class to remain close while pushing them away from embeddings of common classes, thereby promoting greater separability and preserving the distinctiveness of rare conditions in the latent space.

**Cognitive Gap Identification: Discrepancy Between AI and Human Attention**  To quantify the cognitive discrepancy between AI and human reasoning—especially in rare disease cases—we introduce a *cognitive gap loss*. This loss encourages the AI model to attend to features that may be under-utilized by human clinicians, highlighting potential diagnostic blind spots. Formally, we define the loss as:

$$\mathcal{L}_{\text{gap}} = \sum_{i:\, Y_i^{\text{true}} \in \text{rare}} \left\| m_i \odot \nabla_{Z_i} \log p_{\theta_{\text{AI}}}(Y_i^{\text{true}} \mid Z_i) \right\|_2^2,$$

where $Z_i$ is the latent representation, $m_i \in [0,1]^k$ is the learned attention mask approximating human focus, and $\nabla_{Z_i} \log p_{\theta_{\text{AI}}}(Y_i^{\text{true}} \mid Z_i)$ captures the sensitivity of the AI's prediction to each latent feature.

By penalizing high-gradient regions aligned with human attention $m_i$, the model is encouraged to focus on dimensions that are often overlooked, especially in the context of rare diseases. This fosters attentional divergence in rare disease cases, where the AI can uncover atypical patterns that clinicians might miss due to cognitive biases.

## 3.1 TOTAL OBJECTIVE AND TRAINING CURRICULUM

The overall loss function is defined as:

$$\mathcal{L}_{\text{total}} = \mathcal{L}_{\text{rec}} + \mathcal{L}_{\text{AI}} + \mathcal{L}_{\text{human}} + \gamma \mathcal{L}_{\text{contrast}} + \eta \mathcal{L}_{\text{mask}} + \xi \mathcal{L}_{\text{gap}}. \tag{7}$$

The training process follows a staged curriculum, starting with the DAE warm-up using reconstruction loss, followed by the introduction of focal loss for rare disease prediction. The curriculum then adds human cognitive modeling and sparsity regularization, followed by contrastive learning for separating rare and common diseases. Finally, the cognitive gap loss is incorporated to address attention mismatches between AI and human clinicians.

We will train the DAE using the above loss function. Given the learned generative DAE model, we can design the following counterfactual generation tasks.

## 4 COUNTERFACTUAL GENERATION FOR COGNITIVE ANCHORING CORRECTION

To mitigate diagnostic errors from cognitive anchoring, we introduce a counterfactual generation mechanism that leverages the model's probabilistic structure. Given patient data $X$, *if $p_{\theta_{\text{AI}}}$ assigns relatively high probability to a plausible diagnosis $Y_{\text{AI}}$—particularly a rare or under-considered one—that diverges from the human's current diagnosis*, this *triggers counterfactual generation* to challenge the initial decision of human and guide follow-up evaluation or testing.

The **goal** of the counterfactual generation here is to

> *Disrupt doctors' fixation on initial hypotheses by generating alternative diagnostic pathways, particularly for rare diseases.*

**Learning Optimal Perturbation**    The perturbation is learned to increase uncertainty in the human (or human-approximating) model, thus exposing cognitive blind spots.

$$\Delta z^* = \arg \max_{\|\Delta\| \leq \epsilon} \underbrace{\text{Entropy} \left( p_{\theta_{\text{human}}}(m \odot (z + \Delta)) \right)}_{\text{Increase human uncertainty}} \tag{8}$$

Here, $\|\Delta\| \leq \epsilon$ ensures that the changes remain within a medically interpretable range. Without perturbation, the AI's prediction from the original $z$ may align closely with the clinician's current belief. By contrast, perturbing $z$ explores latent variations that introduce diagnostic ambiguity from the human's perspective-potentially uncovering under-recognized or rare conditions.

**Counterfactual Output Generation**    Once the optimal perturbation $\Delta z^*$ is obtained, the system generates two outputs:

- **AI Counterfactual Diagnosis**

$$Y_{\text{cf}}^{\text{AI}} \sim p_{\theta_{\text{AI}}} (z + \Delta z^*) \tag{9}$$

This may yield a rare disease prediction that prompts reconsideration of the original diagnosis.

- **Synthetic Patient Data Generation** An mDAE is used to reconstruct the corresponding patient profile:

$$X_{\text{cf}}' \sim p_\theta (X \mid z + \Delta z^*) \tag{10}$$

Here, $X_{\text{cf}}'$ represents a plausible synthetic patient who presents similarly but includes key missing symptoms supporting the rare disease.

Finally, the system communicates the counterfactual insight as:

> *"Consider alternative diagnoses with similar presentations: [Al-suggested disease $Y_{\mathrm{cf}}^{\mathrm{AI}}$]. If additional findings such as $X_{\mathrm{cf}}'$ were observed, the likelihood of this condition would increase to $p_{\theta_{\mathrm{AI}}}(Y_{\mathrm{cf}}^{\mathrm{AI}} \mid z + \Delta)$."*

This form of explanation aims to encourage the clinician to reflect, reassess, and refine their diagnostic reasoning with evidence-informed support from the AI.

## 5 EXPERIMENT

To evaluate the effectiveness of our proposed framework, we conducted extensive experiments designed to (i) validate robust performance and diagnostic accuracy for *rare disease* detection and (ii) assess the efficacy of counterfactual explanations in addressing cognitive gaps and guiding clinical decision-making.

We used three private, real-world *rare disease* datasets that we constructed and curated in close collaboration with a top-tier hospital, involving multiple departments and clinicians, covering **Gitelman syndrome**, **acromegaly**, and **hypertrophic cardiomyopathy (HCM)**. To support reproducibility, we additionally evaluate on a curated public rare disease dataset for **Granulomatosis with Polyangiitis (GPA)** derived from (Chen et al., 2024). Detailed dataset specifications are provided in Appendix B. Notably, High-quality datasets for rare diseases are scarce, and assembling high-quality rare disease datasets with research value is inherently challenging and constitutes a substantive contribution to the field.

### 5.1 REPRESENTATIONAL AND PREDICTIVE CAPACITY

#### 5.1.1 PREDICTION ON IMBALANCED DATA

The low prevalence of rare diseases leads to imbalanced datasets, posing challenges for conventional classifiers. We present results on a real-world Gitelman syndrome dataset, which has faced difficulties in predicting this disease, with an imbalance ratio of 94:100 to 94:500 (rare disease samples to common disease samples). Standard class-imbalance handling strategies were applied to all baselines, including focal loss for the neural network, SMOTE augmentation for SVM and logistic regression, and built-in imbalance handling for XGBoost and LightGBM.

Our approach outperforms five typical classifiers, as shown in Figure 2, which reports AUC, accuracy for common diseases, and rare disease accuracy. Notably, our model's AUC improves with increasing imbalance, as the larger data volume provides more information for learning despite the greater skew.

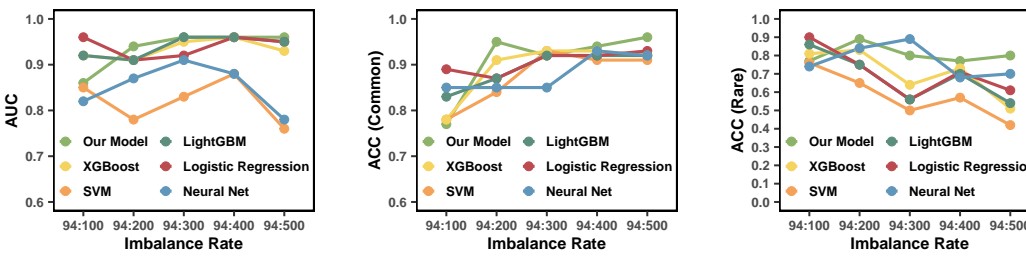

Figure 2: Comparison of model performance under imbalanced data.

#### 5.1.2 LATENT SPACE VISUALIZATION

We visualize the model's latent space using the Gitelman dataset in three distinct ways. These visualizations, shown in Figure. 3, offer valuable insights into the model's internal representations. Panel (a) shows the structural organization of latent embeddings, illustrating the model's ability to encode fine-grained phenotypic details that distinguish clinically similar samples. Panel (b) presents an attention map of clinician focus within the same space: mask values of 1 mark high clinical relevance regions, while 0 indicates lower priority, directly aligning attention with diagnostic importance. Panel (c) highlights features exerting significant influence on human classification decisions, exposing potential decision boundaries where predictions may shift. The visualization

principle involves perturbing latent space vectors to maximize human prediction uncertainty, with the intensity distribution directly reflecting perturbation magnitude. Lighter colors denote higher diagnostic uncertainty, revealing critical knowledge gaps that could lead to misdiagnosis.

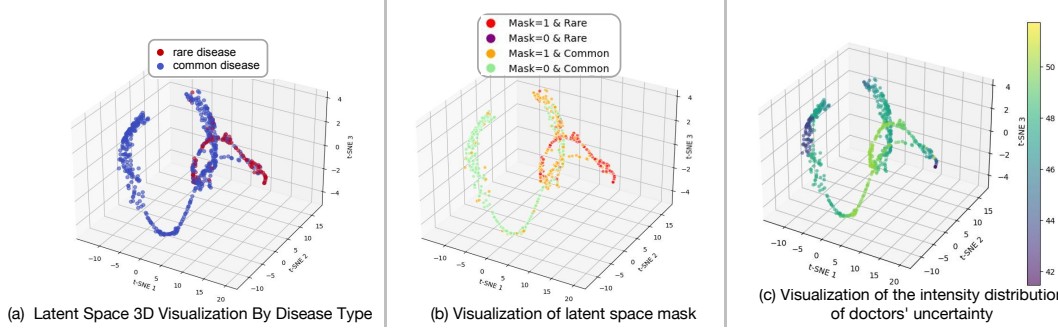

(a) Latent Space 3D Visualization By Disease Type

(b) Visualization of latent space mask

(c) Visualization of the intensity distribution of doctors' uncertainty

Figure 3: Latent space visualization by disease type, clinician attention, and diagnostic uncertainty.

Beyond these views, we also examine how each loss term in fine-tuning stage shapes the latent space via an ablation study in Appendix E. Removing contrastive, gap, or reconstruction loss degrades latent space representation quality, impairing the model's ability to distinguish similar samples; removing prediction or mask-regularization loss degrades AI/human predictors, evidenced by lower AUC and visualizations.

### 5.1.3 AUC ACROSS DATASETS AND ABLATION EXPERIMENTS

We also report AI and human predictors' AUC across 4 datasets (10-run avg ± std, Table 1). The full model shows consistent strong performance, demonstrating robust, stable generalization. Ablations (removing one loss at a time in fine-tuning, stage-wise pretraining unchanged) corroborate each component's necessity: removing AI loss notably degrades AI predictor AUC; removing human loss or mask-regularization loss severely harms human predictors. For specific details regarding the model architecture and hyperparameter selection, refer to F and G.

Table 1: AUC metrics and ablations

|  | Gitelman | | Acromegaly | | HCM | | GPA | |
| --- | --- | --- | --- | --- | --- | --- | --- | --- |
|  | AI | Human | AI | Human | AI | Human | AI | Human |
| Original | **0.96±0.01** | **0.98±0.01** | **0.99±0.01** | **0.98±0.03** | **0.96±0.01** | **0.97±0.01** | **0.88±0.04** | **0.86±0.01** |
| No AI loss | 0.89±0.08 | N/A | 0.96±0.02 | N/A | 0.86±0.05 | N/A | 0.79±0.03 | N/A |
| No human loss | N/A | 0.61±0.14 | N/A | 0.87±0.07 | N/A | 0.73±0.12 | N/A | 0.72±0.02 |
| No mask loss | N/A | 0.90±0.08 | N/A | 0.94±0.05 | N/A | 0.92±0.02 | N/A | 0.81±0.03 |

### 5.2 LLM-HUMAN DUAL QUANTITATIVE EVALUATION OF COUNTERFACTUALS

To address the challenges and diagnostic needs in rare disease medicine, our model supports counterfactual analysis across diverse scenarios. We constructed three representative counterfactual scenarios: **Scenario 1**: Feature Completion for Low-Confidence Predictions, **Scenario 2**: AI-Human Prediction Discrepancy Resolution, and **Scenario 3**: Uncertainty-Driven Alternative Diagnoses. For detailed descriptions of these scenarios, please refer to Appendix D.

For a more comprehensive assessment, an LLM- and doctor-based evaluation framework is designed for evaluating counterfactual outcomes. Fig. 4 shows Prompt, LLM evaluations and doctor evaluations across three scenarios. For LLM prompting specifics and responses, See Appendix H.

**Evaluated by LLM**  Since 2023, LLMs with advanced instruction-following and semantic comprehension have enabled automated evaluation (Gao et al., 2025). In our framework, pre-trained LLMs assess counterfactuals using structured prompts, evaluating plausibility, relevance, and cognitive support across semantic, causal, and operational dimensions..

**Evaluated by Doctors**    Clinical experts from a leading hospital validated rare disease counterfactuals for medical plausibility and clinical relevance, leveraging their domain expertise.

Figure 4: Illustration of prompt, LLM response segment and real world doctor evaluation segment.

## 5.3 QUANTITATIVE EVALUATION OF COUNTERFACTUALS AGAINST BASELINES

**Baselines**    We compare our method with two baseline approaches: REVISE (Joshi et al., 2019), which uses optimization within a generative model's latent space, and CF-VAE (Nagesh et al., 2023), which optimizes a variational autoencoder alongside a binary prediction model.

**Metrics**    We evaluate counterfactuals based on two metrics: (1) **Label Flip Rate**: The proportion of counterfactuals correctly classified into the target class, indicating validity. (2) **Root Mean Squared Error (RMSE)**: Measures the perturbation magnitude between the counterfactual and original input, with lower RMSE indicating higher plausibility.

**Results**    Table 2 compares our model, REVISE, CF-VAE, and an ablation experiment across four datasets. Our model achieves the highest label flip rate and lowest RMSE, outperforming all baselines in generating valid and minimally perturbed counterfactuals.

Table 2: Performance metrics across four datasets.

| Model | Gitelman | | Acromegaly | | HCM | | GPA | |
|---|---|---|---|---|---|---|---|---|
| | Label Flip Rate | RMSE | Label Flip Rate | RMSE | Label Flip Rate | RMSE | Label Flip Rate | RMSE |
| REVISE | 0.96±0.03 | 5.40±0.89 | 0.92±0.11 | 13.96±14.44 | 0.70±0.40 | 0.33±0.04 | 0.94±0.07 | 0.18±0.06 |
| CFVAE | 0.96±0.02 | 12.00±1.77 | 0.85±0.15 | 13.96±14.84 | 0.80±0.40 | 0.33±0.01 | 0.85±0.18 | 0.29±0.09 |
| Our Model | **1.00±0.00** | **1.93±0.76** | **1.00±0.00** | **0.18±0.10** | **1.00±0.00** | **0.10±0.13** | **1.00±0.00** | **0.12±0.03** |
| Ablation | **1.00±0.00** | 4.85±3.27 | **1.00±0.00** | 0.21±0.08 | **1.00±0.00** | 0.46±0.27 | **1.00±0.00** | 0.25±0.07 |

## 6 CONCLUSION

We introduced a human-centered counterfactual reasoning framework that perturbs latent patient representations via a DAE-based latent state generative model to counter cognitive anchoring in rare disease diagnosis. By generating realistic "what-if" scenarios, our method surfaces overlooked conditions and guides clinicians toward alternative hypotheses. A mixed LLM- and doctor-based evaluation confirms the scientific soundness and clinical relevance of the generated cases. This framework fosters reflective diagnostic reasoning, enhances interpretability, and offers a scalable tool for bridging human knowledge gaps in challenging medical scenarios.

REPRODUCIBILITY STATEMENT

If accepted, all codes and the public dataset used in this work will be made publicly available.

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

## A    RELATED WORK

**Counterfactual Explanations**    The evolution of counterfactual explanations has transitioned from optimizing feature perturbations (Wachter et al., 2017) to frameworks that prioritize human-AI collaboration and safety. Early methods focused on generating minimal feasible changes (e.g., DiCE (Mothilal et al., 2020)), but were criticized for ignoring user-specific constraints and real-world applicability (Verma et al., 2020). More recent work, including (Lee & Chew, 2023), highlights the role of counterfactuals in mitigating cognitive biases. (Lee & Chew, 2023) showed that exposing users to hypothetical scenarios reduces overreliance on erroneous AI predictions, particularly among non-experts susceptible to confirmation bias. This aligns with broader findings in human-AI interaction, where explanations must balance interpretability with decision accuracy (Buçinca et al., 2021; Straitouri et al., 2024). A significant advancement in this area is the formalization of counterfactual harm, defined as the risk that explanations may degrade human judgment. (Straitouri et al., 2024) introduced structural causal models with conformal risk control to bound harmful outcomes in clinical systems. Their approach integrates monotonicity assumptions (e.g., "higher biomarker values correlate with worse prognosis") to ensure explanations align with domain knowledge, thereby addressing a gap in earlier optimization-based methods (Van Looveren & Klaise, 2021). This shift reflects a growing emphasis on safety-critical metrics, moving beyond traditional criteria like sparsity and realism (Verma et al., 2020).

**Counterfactual Generative Models**    Generative models have been introduced to generate numerical counterfactuals, enabling dynamic adaptation to user constraints. Early GAN-based approaches, such as CounterRGAN (Nemirovsky et al., 2022), enforced immutable features via residual networks but lacked flexibility for real-time customization. FCEGAN (Hellemans et al., 2025) addresses this limitation by incorporating user-defined templates and dual discriminator losses, facilitating personalized explanations in domains like loan approvals (Yang et al., 2022). These frameworks align with CTGAN's training-by-sampling strategy (Xu et al., 2019) to handle class imbalance, a persistent challenge in financial and medical datasets. While REVISE (Joshi et al., 2019) introduced a method for generating numerical counterfactuals using arbitrary generative models, it can produce unrealistic counterfactuals, making them unsuitable for healthcare applications, and is limited by the need for multiple calls to an optimization module. Although CFVAE (Nagesh et al., 2023) was designed for generating counterfactuals in healthcare settings using variational autoencoders, it does not account for realistic challenges in healthcare, such as class imbalance in rare disease cases and missing values in datasets. To overcome these limitations, we propose a novel method designed for healthcare applications, particularly in rare disease diagnosis. Our approach generates personalized counterfactuals for clinicians while handling missing values and class imbalance in the training data.

## B    EXPERIMENTAL DATASETS

To evaluate our method, we consider the following three private datasets.

**Gitelman Syndrome** This dataset comprises real clinical records from a top hospital, focusing on Gitelman syndrome (GS), a rare autosomal recessive renal tubulopathy. The data contains 594 patients, including 94 diagnosed with GS and 500 non-GS individuals. Five key diagnostic features are included: *Serum Potassium*, *Urine Potassium*, *pH*, *Bicarbonate*, and *High Blood Pressure*, with labels derived from clinical diagnoses. To emulate real-world scenarios where critical test results are missing (a common challenge in rare disease diagnosis), we retain the missing values in the original data. This enables counterfactual analysis to quantify how missing tests impact predictions, thereby guiding clinicians to prioritize specific examinations for undiagnosed cases. The dataset is split into 80%-20% train-test sets for GS classification, with subsequent counterfactual perturbation analysis performed in the latent space of the complete data. It should be noted that we retained the situation of data imbalance, which is to be consistent with the situation that the incidence of rare diseases in the real world is much lower. And despite this imbalance, our model still maintained good performance.

**Acromegaly** This dataset includes real-world clinical records from a top hospital, focusing on acromegaly, a chronic disorder caused by excessive growth hormone (GH) secretion, typically due to pituitary somatotroph adenomas. The data contains 181 patients, comprising 88 diagnosed with acromegaly and 93 non-acromegaly controls. Three clinically significant features are incorporated: *Serum GH*, *IGH-1*, and *OGTT-GH_min*, with labels derived from clinical diagnoses. To reflect realistic

data incompleteness, we retain naturally occurring missing values in the original dataset and explicitly record their positions. This facilitates counterfactual generation that aligns with clinical practice, allowing clinicians to evaluate how incomplete laboratory profiles influence diagnostic predictions. The dataset is partitioned into 80%-20% training-test sets for binary classification, followed by counterfactual perturbation and interpretability analysis in the latent space of the complete data to identify critical diagnostic drivers.

**Hypertrophic Cardiomyopathy (HCM)** This dataset includes real-world clinical records from a top hospital, focusing on hypertrophic cardiomyopathy (HCM), an inherited cardiac disorder characterized by abnormal myocardial thickening that may lead to ventricular outflow tract obstruction, arrhythmias, and heart failure. The data contains 36 patients, including 21 HCM-diagnosed individuals and 15 individuals with another rare disease (ATTR, amyloidosis trans-thyretin related) as the control group. Eight clinically significant features are incorporated: *Asymmetric Hypertrophy*, *SAM*, *Low Left Ventricular Voltage*, *High Left Ventricular Voltage*, *Family History*, *Sarcomere Gene Mutation*, *TTR Gene Mutation*, and *Amyloid Deposition*. Similarly, to preserve clinical authenticity, naturally occurring missing values in the original dataset are retained and explicitly mapped for interpretability. The dataset is partitioned into 80%-20% training-test splits for HCM classification. Post-training, counterfactual perturbation and causal analysis are conducted in the latent space of the complete data to identify critical diagnostic patterns and feature interactions.

**Granulomatosis with Polyangiitis (GPA)** This dataset, relevant to the context of the file s13023-019-1040-6.pdf, is derived from (Chen et al., 2024) and contains real-world clinical records targeting granulomatosis with polyangiitis (GPA), which is an ANCA-associated vasculitis frequently linked to PR3-ANCA and upper-airway/pulmonary involvement. The cohort includes 93 subjects in total, comprising 11 patients diagnosed with GPA and 82 non-GPA controls that are deliberately selected for their high clinical confusability with GPA in ENT (ear, nose, and throat) and respiratory presentations; it incorporates seven clinically meaningful binary features, namely *Otitis Media*, *Hemoptysis*, *Proteinase 3 Antibody Titer, Elevated (PR3-ANCA)*, *Cytoplasmic ANCA (c-ANCA) Present*, *Knee Pain (Bilateral)*, *Peripheral Cyanosis (one month to one year)*, and *Rhinitis*, with labels derived from clinical diagnoses. The dataset is split into an 80%-20% training–test partition for binary GPA classification, with naturally occurring missing values retained and their positions mapped; subsequent to the classification task, counterfactual perturbation and interpretability analysis are conducted in the latent space of the completed data to identify key diagnostic drivers and interactions between symptoms and serological indicators.

## C RATIONALE FOR CHOOSING MASKED DENOISING AUTOENCODER (MDAE): A COMPARISON WITH OTHER AUTOENCODER VARIANTS

To clarify why the Masked Denoising Autoencoder (mDAE) is selected for our framework rather than other autoencoder variants, we conduct a comparative analysis of Deterministic Autoencoder (AE), Variational Autoencoder (VAE), Denoising Autoencoder (DAE), and their derivatives, with a focus on their suitability for rare disease diagnostic scenarios:

**Deterministic Autoencoder (AE)** AEs lack mechanisms to handle noise or missing values, which are common in rare disease data. They overfit to sparse inputs, producing unreliable latent representations and counterfactuals, making them unsuitable.

**Variational Autoencoder (VAE)** VAEs, as generative models, center on modeling the joint distribution information. Furthermore, their inherent stochasticity in latent spaces hinders precise, targeted counterfactuals needed to correct cognitive anchoring, often generating implausible clinical values and propagating biases via heuristic imputation of missing data, which limits clinical utility.

**Denoising Autoencoder (DAE) and Masked Denoising Autoencoder (mDAE)** DAEs are explicitly designed to process corrupted or incomplete inputs, with a focus on modeling $p(\text{missing } \mathbf{x} \mid \text{observed } \mathbf{x})$. Standard DAEs enhance noise robustness but lack dedicated handling of missing data (a common challenge in rare disease), **a gap our masked DAE (mDAE) fills** by explicitly training on partially observed data via sparse masks to reconstruct complete profiles. It generates deterministic

latents for precise control over counterfactual perturbations and enforces physiological constraints to ensure clinically valid outputs in rare disease scenarios.

## D    COUNTERFACTUAL SAMPLE GENERATION

To address the challenges and diagnostic needs in rare disease medicine, our model supports counterfactual analysis across diverse scenarios. In this section, we present three representative and practically relevant scenarios for detailed evaluation.

**Scenario 1: Feature Completion for Low-Confidence Predictions**: When a patient's original input features have missing values, overlap significantly with common disease characteristics, and yield low-confidence AI predictions for common diagnoses, our model generates counterfactual samples to address missing features. This refines clinical judgments and guides decision-making.

**Scenario 2: AI-Human Prediction Discrepancy Resolution**: In situations where AI predictions diverge from clinician diagnoses, our model produces counterfactual "flipped" samples to highlight the underlying differences in decision-making logic. These samples provide interpretable evidence that helps clinicians reconcile inconsistent conclusions.

**Scenario 3: Uncertainty-Driven Alternative Diagnoses**: By perturbing feature vectors in latent spaces where clinicians exhibit maximal diagnostic uncertainty, our model generates alternative diagnosis lists. This anchors cognitive bias correction and supports robust differential diagnosis.

## E    LATENT SPACE VISUALIZATION WITH ABLATION STUDY

We conduct an ablation study to evaluate the necessity of each loss term in our model's total loss function. Specifically, we visualize the distribution of the latent space when individually removing each loss component during fine-tuning (prior to fine-tuning, each component of our model, including DAE, AI predictor, mask net and human predictor, is first trained in stages with its corresponding loss function). As shown in Figure. 5, Our findings indicate that the removal of the contrastive loss, gap loss, or reconstruction loss degrades the quality of the latent space representation, thereby impairing the model's ability to discriminate between similar samples.

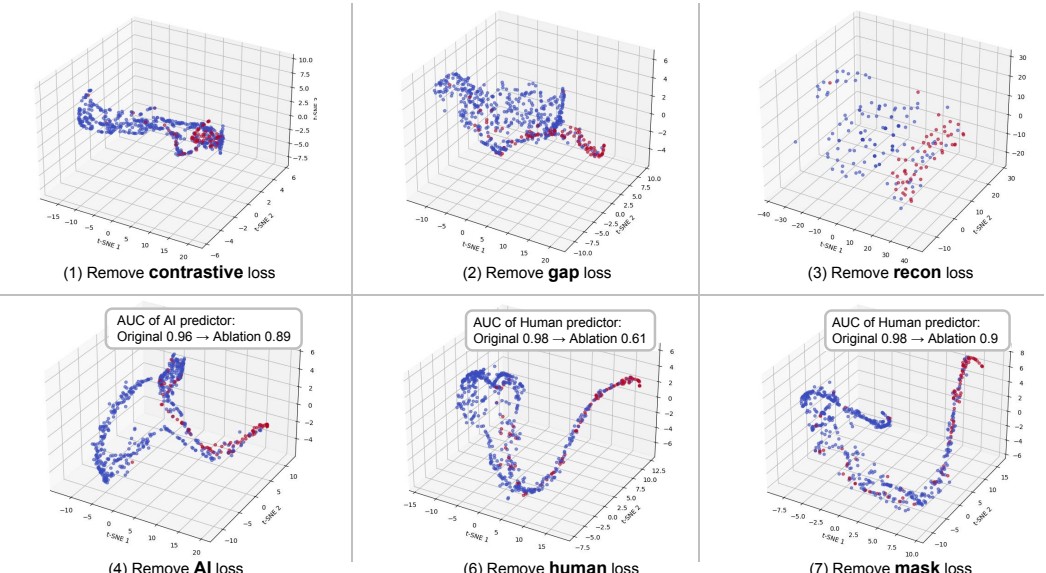

Figure 5: Ablation study: loss function removal impact on latent space and model performance.

In contrast, removal of the AI prediction loss, AI prediction loss or mask regularization loss impairs the performance of the AI predictor or human predictor, as depicted by the AUC changes in the figure, underscoring the indispensable role of each loss component in maintaining model effectiveness.

# F   MODEL ARCHITECTURE DETAILS

## F.1   DAE ARCHITECTURES

The Denoising Autoencoder (DAE) architecture captures clinical feature mappings through an Encoder and Decoder. The Encoder uses ELU activations to project raw features into a 32-dimensional latent space, while the Decoder reconstructs inputs from this space. Categorical features are embedded via a dedicated layer, and the design supports robust learning from incomplete data. Take the Gitelman syndrome dataset as an example, key components are detailed in Table 3, which outlines layer dimensions and functional roles.

Table 3: DAE architecture configuration

| Component | Layers | Dimension | Functional Description |
|---|---|---|---|
| Encoder | Input Layer | 5 | Raw clinical features |
| | Hidden Layer | 128 | ELU-activated transformation: $h = \text{ELU}(Wx + b)$ |
| | Latent Space | 32 | Bottleneck representation: $z$ |
| | Embedding | 8 | Categorical feature encoding: $\text{onehot}(x)W_e$ |
| Decoder | Input Layer | 32 | Latent space input: $z$ |
| | Hidden Layer | 128 | Feature decoding: $h_d = \text{ELU}(W_d z + b_d)$ |
| | Output Layer | 5 | Feature reconstruction: $\hat{x}$ |

## F.2   PREDICTOR ARCHITECTURES

The AI and human predictors, along with the attention mask network, are designed to explicitly model the divergence between machine and clinician reasoning. The AI predictor operates in the full latent space to generate ground truth-aligned diagnoses, while the human predictor uses a sparse attention mask (generated by the mask network) to simulate cognitive constraints in clinical decision making. Table 4 outlines the architecture details, including layer dimensions, activation functions, and the attention mechanisms. This modular design supports interpretable counterfactual generation by isolating human-AI cognitive gaps in the latent space.

Table 4: Predictor Architectures Configuration

| Component | Layers | Dim/Num of Heads | Description |
|---|---|---|---|
| AI Predictor | Input Layer | 32 | ELU-activated projection into hidden space |
| | Hidden Layer | 128 | ELU transformation of latent features |
| | Output Layer | 2 | Produces class logits for prediction |
| Mask Network | Input Layer | 5 | ELU-activated linear embedding |
| | Attention Layer | 4 | Multi-head self-attention for contextual feature interaction |
| | Output Layer | 32 | Generates masking coefficients |
| Human Predictor | Input Layer | 32 | Takes the masked latent representation as input |
| | Hidden Layer | 128 | ELU transformation of masked latent space |
| | Output Layer | 2 | Produces class logits aligned with experts |

# G   TRAINING CONFIGURATION DETAILS

## G.1   HYPERPARAMETER AND LOSS WEIGHT SELECTION

All hyperparameters and loss weights were selected via a systematic grid search confined strictly to the training set, ensuring that the independent 20% test set remained untouched throughout model development and thereby preventing data leakage.

Within the 80% training set, we adopted a hold-out validation strategy: 70% of the data were used for model fitting, and the remaining 30% served as a validation subset to evaluate hyperparameter configurations.

- **Learning rate** was searched over the range $[10^{-5}, 10^{-3}]$.

- **Most loss function weights** were searched over a clinically relevant range of $[0.1, 2.0]$.

- **Mask sparsity loss weight**, due to its role as a regularization term requiring finer control to balance sparsity constraints and model performance, was searched over the narrower range of $[10^{-5}, 0.1]$.

Searches were guided by validation AUC, with priority given to configurations demonstrating stable performance (AUC variance $< 0.02$) across three random seeds. The final hyperparameters and loss weights were chosen based on the best validation AUC while ensuring model outputs remained within clinically plausible ranges.

## G.2 STAGE-WISE TRAINING DETAILS

The model is trained in four stages: DAE warm-up, AI predictor training, joint human predictor and mask network training, and fine-tuning. Table 5 specifies the learning rate schedules, batch sizes, and regularization strategies (e.g., gradient clipping) for each phase on the Gitelman syndrome dataset. For instance, the DAE warm-up phase employs learning rate annealing and early stopping to stabilize latent space initialization. This staged approach balances model complexity and training stability while ensuring task-specific optimization.

Table 5: Progressive training strategy

| Phase | Components | Learning Rate | Key Details |
|---|---|---|---|
| DAE Train | Encoder / Decoder | 1e-4 | • LR annealing
• Early stop
• Gradient clip $\leq 1.0$
• Batch size 16 |
| AI Predictor Train | AI Predictor Network | 1e-4 | • LR annealing
• Early stop
• Gradient clip $\leq 1.0$
• Batch size 16 |
| Human Predictor + Mask Net Train | Human Predictor Network, Mask Network | 1e-4 | • LR annealing
• Early stop
• Gradient clip $\leq 1.0$
• Batch size 16 |
| Fine-Tuning | Full Network | 1e-4 | • Gradient clip $\leq 1.0$
• Batch size 16 |

## G.3 LOSS FUNCTION WEIGHT IN FINE-TUNING STAGE

The total training loss combines multiple objectives, including reconstruction, classification, contrastive separation, and cognitive gap minimization. Table 6 defines the weights assigned to each loss component on the Gitelman syndrome dataset, emphasizing the balance between feature reconstruction (dominant in early stages) and rare/common disease separability (enforced via contrastive loss).

Table 6: Loss Function Specification

| Loss Type | Weight | Function |
|---|---|---|
| Reconstruction | 1 | Reconstruct input features |
| AI | 1 | Maximize AI prediction accuracy |
| Human | 1 | Align with human diagnoses |
| Mask | 0.001 | Promote sparse attention masks masks |
| Contrastive | 1.5 | Separate rare/common diseases |
| Gap | 1.5 | Reduce human-AI attention gaps |

## H  DETAILS OF PROMPTING LLM AND COUNTERFACTUAL EVALUATIONS

Figure. 6 illustrates the operational mechanism of prompting the LLM and LLM response across three counterfactual scenarios. For each scenario, a representative case is selected: the first from the acromegaly dataset, and the latter two from the Gitelman dataset. This visual depiction not only offers profound insights into the framework's functionality but also provides a practical reference for clinicians and researchers, underscoring the significance of counterfactual reasoning in enhancing the differential diagnosis of rare diseases.

## I  BROADER IMPACT AND LIMITATION

This study aims to address the underdiagnosis of rare diseases caused by cognitive biases in clinical decision-making. Our framework helps clinicians consider rare conditions more effectively through generative counterfactuals, potentially reducing diagnostic delays and improving patient outcomes, especially in underserved areas with limited specialized expertise. By modeling the cognitive gaps between humans and AI, it promotes transparent and bias-aware collaboration, setting a practical example for AI applications in healthcare and other high-stakes fields. Potential risks include the possibility of over-relying on AI, which we mitigate by designing interpretable counterfactual explanations to supplement, rather than replace, clinical judgment.

One key limitation of this study stems from the long-standing challenge of rare disease data acquisition and sharing, a core bottleneck in rare disease research. Unlike common disease domains with accessible public datasets, there is a severe scarcity of open, high-quality data resources for rare diseases. To address this gap, our team invested over a year in close collaboration with a top-tier hospital, engaging multiple departments and relying on substantial clinician effort—to construct three private rare disease datasets tailored to this research. However, due to strict ethical and privacy constraints, these datasets cannot be fully made public. To alleviate this limitation in the future, we aim to develop and release a rare disease data simulator: this tool will generate synthetic data with characteristics consistent with real rare disease cases, supporting the reproducibility of related research while upholding privacy protection requirements.

## J  THE USE OF LARGE LANGUAGE MODELS (LLMS)

Large language models (LLMs) were used in this work exclusively for polishing the writing and correcting grammar errors. All substantive research ideas, methodological design, and scientific conclusions presented in this paper were independently developed and validated by the authors.

## K  ACKNOWLEDGMENTS

We would like to thank the doctors from various departments of the Top Hospital for their efforts in providing the rare disease dataset, and for evaluating and offering valuable suggestions on our counterfactual results.

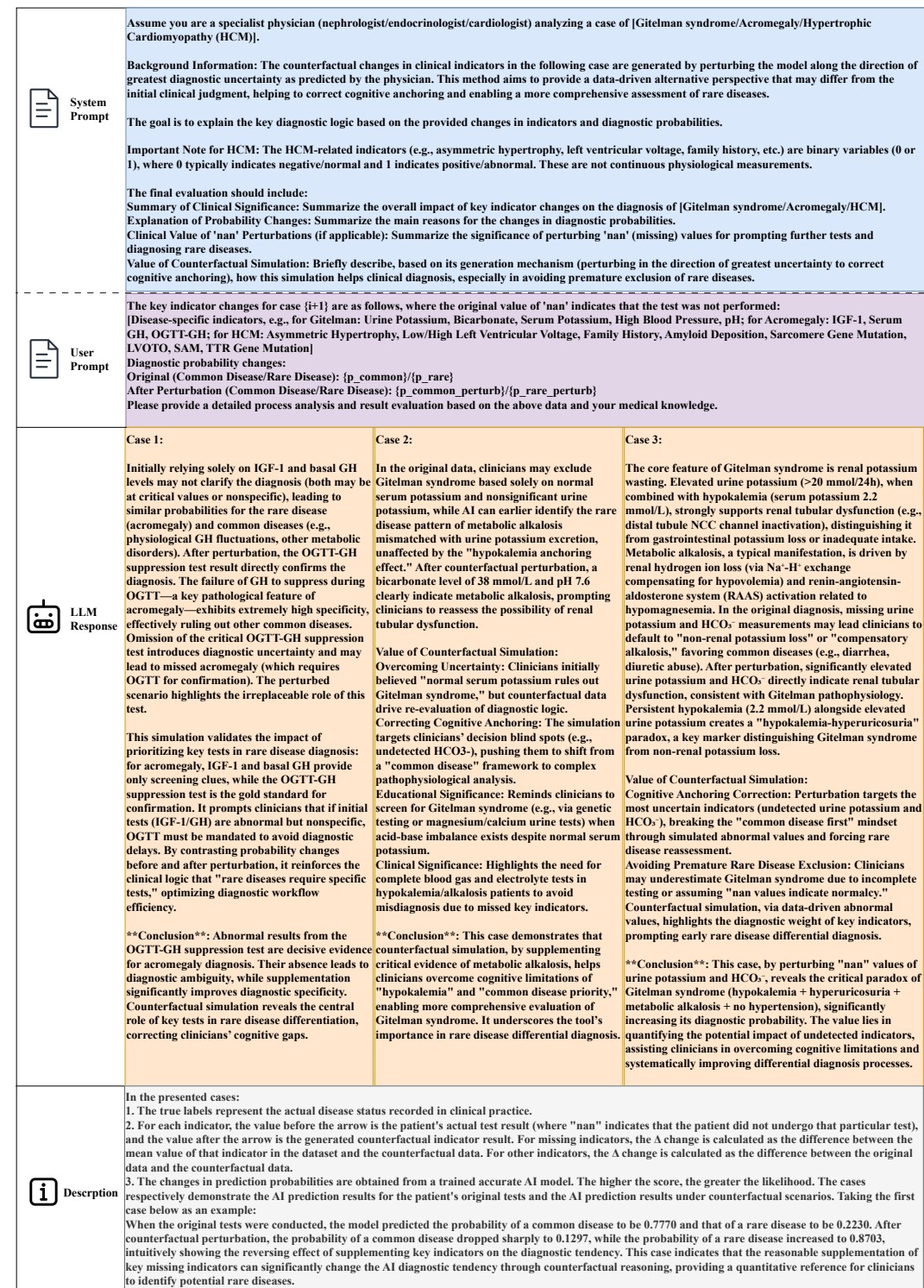

Figure 6: Prompting LLM and LLM response under three counterfactual scenarios

## L Computing Infrastructure

All synthetic data experiments are performed on Ubuntu 20.04.3 LTS system with Intel(R) Xeon(R) Gold 6248R CPU @ 3.00GHz, 227 Gigabyte memory.

