# OpenReview forum: "Unanchoring the Mind: AI-Guided Counterfactual Reasoning for Rare Disease Diagnosis"
_ICLR.cc/2026/Conference — ICLR 2026 Conference Withdrawn Submission_

### Official Review · Reviewer_6P3K · 2025-10-29

**Soundness:** 2
**Presentation:** 2
**Contribution:** 2
**Rating:** 2
**Confidence:** 2

**Summary:**

This paper proposed a denoising autoencoder (DAE) to address cognitive anchoring for human clinician diagnosis thought perturbing latent patient space to generate contrastive counterfactuals about potential rare diseases. This paper has tested the proposed methods on three private datasets and one public rare disease dataset.

**Strengths:**

- This paper points out an interesting cognitive anchoring behavior for human clinician diagnosis.
- It has contextualized the cognitive anchoring for rare disease diagnosis that they share similar and common symptoms.
- It tests the proposed methods on three private datasets and one public rare disease dataset.

**Weaknesses:**

- It is unclear about the effectiveness of the propose methods, even it outperforms on AUC score, but there might exist high false positives for rare disease prediction, how to reduce false positives can be a huge challenge. This paper did not provide details to discuss false positive issues.
- This paper did not choose the fair baseline methods, because it uses contrastive learning and several other loss functions, it should make sense to compare with the same prediction methods using contrastive learning.
- The human evaluation looks vague and general, did not have rigorous rubric to measure the effectiveness and difference.

**Questions:**

- can you provide more details about the false positive cases and how you are going to address false positive cases for your prediction.
- when you provide counterfactuals, are these counterfactuals similar for one specific rare disease?
- evaluation is critical, especially clinical evaluation, clinical soundness, please conduct rigorous clinical evaluation.

---

### Official Review · Reviewer_Yqjw · 2025-10-30

**Soundness:** 2
**Presentation:** 3
**Contribution:** 2
**Rating:** 2
**Confidence:** 3

**Summary:**

This paper proposes a DAE-based generative framework to combat cognitive anchoring in rare disease diagnosis. The model is designed to generate counterfactuals that challenge a clinician's initial assumptions. Its main novelty lies in a dual-predictor architecture that jointly learns to predict the ground-truth diagnosis (AI predictor) and a clinician's likely diagnosis (human predictor) from a shared latent space. The "cognitive gap" between these two is explicitly modeled via a sparse attention mechanism and a dedicated loss term. Counterfactuals are generated by perturbing the latent representation in a direction that maximizes the diagnostic uncertainty of the simulated human model, thereby surfacing rare but plausible alternative diagnoses. The authors evaluate their method on one public and three private datasets, using a combination of ML metrics, LLM-based assessment, and expert clinician feedback to demonstrate its effectiveness.

**Strengths:**

1. The paper tackles a well-recognized and highly impactful problem in clinical medicine—cognitive bias in diagnosis. The framing of the solution as a human-AI collaborative tool to "unanchor the mind" is compelling and conceptually sophisticated.

2. The central idea of a dual-predictor model that explicitly simulates a human's diagnostic process to identify and target cognitive gaps is highly innovative. The counterfactual generation mechanism, which optimizes for the human model's uncertainty, is elegant and psychologically well-motivated.

**Weaknesses:**

1. Reproducibility Issues: The paper's empirical claims are built almost entirely on private datasets (3 out of 4). For a top-tier machine learning conference, where verifiable and reproducible results are paramount The contribution of curating the data is noted, but it does not absolve the authors of the responsibility to provide a basis for the community to verify their claims. The promise of releasing a data simulator in the future is irrelevant to the evaluation of the current submission. Without access to the data, the impressive results in Figures 2 and Table 2 are essentially unverifiable and rarely meaningful to the community.

2. The entire framework hinges on the model's ability to simulate a "human doctor's diagnosis" (Y_human). However, this is based on "the clinician’s recorded label". This is a gross oversimplification that borders on being conceptually unsound. There is no single "human" model of diagnosis. This label could be from a world-class expert or a junior resident; it could be an initial guess or a post-consultation final diagnosis. The model is not learning "human cognitive bias"; it is learning to mimic the labeling behavior of a small, specific, and undefined group of clinicians present in a private dataset. This undermines the central scientific claim of the paper.

3. Unscientific Evaluation of Counterfactuals: The paper presents a "LLM-Human Dual Quantitative Evaluation," but this evaluation is weak and lacks rigor.

LLM Evaluation: Using an LLM to "evaluate" clinical plausibility is not a scientifically valid method. LLMs are known to hallucinate and produce text that is plausible but factually incorrect, especially in expert domains. The prompt in Figure 6 is also highly leading, effectively telling the model the desired outcome. This part of the evaluation is, at best, an interesting anecdote and, at worst, scientific theater.

Human Evaluation: The evaluation by "clinical experts" is presented without any necessary detail. How many experts? What was their specialty? What was the exact protocol? Were they blinded to the AI's original prediction? What was the inter-rater reliability? Without these details, the claim of "doctor-based evaluation" is unsubstantiated.

4. Overly Complex and Brittle Model: The final loss function (Eq. 7) is a weighted sum of six distinct terms. This introduces at least five hyperparameters that need careful tuning. While the authors describe a grid search, such complexity raises serious concerns about the model's brittleness and the generalizability of the reported results. It is unclear if this performance can be achieved on a new dataset without an exhaustive and potentially dataset-specific tuning process.

**Questions:**

1. The validity of your entire framework rests on the Y_human variable. Can you provide a precise definition of this label? Specifically: who provided the label (level of expertise), at what stage of the diagnostic process was it recorded (initial vs. final), and how do you justify that learning from these specific labels allows you to model a generalizable "human cognitive bias"?

2. Your claim of clinical utility relies on expert evaluation. Can you provide the full protocol for this evaluation, including the number of reviewers, their qualifications, the instructions they were given, and the quantitative results (e.g., scoring, inter-rater agreement)?

3. The non-reproducibility of this work is a critical issue. Given that 75% of your datasets are private, how can the scientific community trust or build upon your results? Why should this paper be accepted at ICLR without the possibility of independent verification?

4. The optimization in Eq. 8 to find \theta z* is central to your method. This is a non-concave optimization problem. Can you describe the algorithm used, its stability, and how sensitive the quality of the generated counterfactuals is to the hyperparameters of this optimization (e.g., step size, number of steps, \epsilon)?

---

### Official Review · Reviewer_DxeW · 2025-10-31

**Soundness:** 2
**Presentation:** 3
**Contribution:** 2
**Rating:** 4
**Confidence:** 3

**Summary:**

The paper proposes a counterfactual framework that adopts a Denoising Autoencoder (DAE). The jointly learns the distribution of true diagnostic labels and human diagnostic behavior. The proposed framework generates counterfactuals that highlight rare but plausible conditions, whereby in some cases they may be clinically ignored, but plausible.

**Strengths:**

•	The real world evaluation is particularly valuable.
•	The ability for the propose DAE approach as to existing works based on AE/VAE ensures handling of noisy/incomplete data, this is particularly common in medical data.
•	The idea itself, to my knowledge is novel. The combination of a human predictor and AI predictor seems valuable to incorporate a cognitive aspect of counterfactual generation.
•	The quantitative results favour the proposed method.

**Weaknesses:**

•	The definition of the human label seems mildly ambiguous, as there is little information about the collection of the label. The setting of the label would be critical to the effectiveness of the proposed framework. Thus, understanding when to best collect data to facilitate the framework would be ideal.
•	Cognitive anchoring is a strong assumption. One would think that humans can observe what is not contained in the data. For example a medical expert may observe physical reactions or patient well-being. This would not be a form of bias, but instead useful observables. It is unclear how the mitigation of these considerations would be addressed – consequently, I think actionability is something that needs to be considered in such method, whereas this is not discussed.
•	Selection of the baseline hyperparameters is unclear.
•	There is no clear evaluation against counterfactual desiderata. Whilst validity and proximity (I’m unsure if plausibility is the correct term here) are implicit to the evaluation, a broader spectrum of desiderata are not discussed in detail.

**Questions:**

How did the authors select hyperparameters for competing benchmarks (e.g. $\lambda_S$ and $\lambda_{cf}$ as per the CF-VAE paper)?
At what point was the human label collected within these datasets? (e.g. initial vs final diagnosis)
The quality of this label seems fundamental to the effectiveness of the approach. Do the authors have any guidance or recommendation on in medical settings, which point this label should be collected – is there a guidance on collection to make best use of the approach in real-world deployment?
Could the authors please expand on the rationale for maximising human uncertainty?
Is actionability / actionable guidance considered in counterfactual generation? Aligning with the weakness of assumed cognitive anchoring, whereas it could be observational details that sway a medical professionals’ decisions.

---

### Official Review · Reviewer_cxgP · 2025-11-02

**Soundness:** 2
**Presentation:** 3
**Contribution:** 3
**Rating:** 4
**Confidence:** 3

**Summary:**

The paper proposes a masked denoising autoencoder (mDAE) that learns a 32-dimensional latent space from partially observed clinical feature vectors and augments it with (i) a sparsity-regularized “human” head guided by a learned mask m, (ii) an AI prediction head for
rare/common disease discrimination, (iii) a contrastive loss to keep rare-disease latents distinct from common-disease latents, and (iv) a “cognitive gap” loss that penalizes gradients aligned with the human attention mask so the AI attends to evidence clinicians may
under-use. Counterfactual samples are then generated in latent space for three scenarios: completion of missing features, resolving AI-human disagreement, and proposing alternatives under high human uncertainty. Experiments on four rare-disease datasets (Gitelman, Acromegaly, HCM, GPA) include 10-run mean±std AUCs and ablation assessments removing individual loss terms; a figure explores class-imbalance robustness versus baselines; and a dual evaluation of counterfactuals by an LLM and clinicians is presented.

**Strengths:**

The paper targets an important and not very studied setting (rare-disease diagnosis with missing data and cognitive anchoring) and proposes a conceptually coherent system that couples an imputation-capable mDAE with losses that are each motivated by the task: an L1
mask promotes sparse “human” attention consistent with limited cognitive bandwidth; a contrastive term counteracts rare-class collapse; and the cognitive-gap penalty operationalizes the aim of nudging the AI toward evidence clinicians underweight. The method considers precise loss definitions and an explicit staged curriculum, making reproduction plausible. The empirical section reports multi-dataset 10-run AUCs with ablation results that show substantial performance decrease when removing the human/mask or AI losses, as an evidence that the
full objective is needed. The paper also presents class-imbalance stress tests and clear latent-space visualizations add qualitative intuition
that complements the metrics. The evaluation of counterfactuals using both clinicians and an LLM is an interesting and apparently effective strategy, since it takes into account cost constraints while still considering human judgment

**Weaknesses:**

The performance evaluation is based on AUCs, but considering that we are talking about rare-events, AUC/accuracy may be
insensitive to low prevalence. For instance, the paper does not report PR-AUC or calibration as suggested by related work (Juba & Le, 2019), which weakens some performance claims. The statement that lower RMSE implies higher plausibility is not really demonstrated, since RMSE is a proximity measure, sensitive to feature scaling/types, and it does not consider basic clinical constraints (ranges, binarity, thresholds/monotonicities). Counterfactual assessment needs to be improved because distance and label flips does not seem to be enough to evaluate its robustness and faithfulness. It is not clear which LLM was used, as well as its associated settings, and clinician evaluation relevant details are not provided, such as number of professionals per case, blinding, and inter-rater agreement. As it is, it is not clear to what extent the paper is reproducible and the validity of human evaluation. The paper also claims that the gap loss enable the AI system to consider features that clinicians do not employ frequently , but there is no direct demonstration of an off-mask shift (e.g., change in
feature importance/gradients relative to the mask) or of reduced anchoring in behavior. Further, ablations presented just ground the need for classification. The paper should improve its statistical analysis, which is currently limited to reporting mean +/- SD across runs without confidence intervals or simple significance tests vs. baselines. Finally, the HCM dataset is quite small and the control group consists only of patients with another specific rare disease, limiting the validity of the results.

**Questions:**

- Could you report PR-AUC and calibration metrics in addition to AUC, and discuss how results change across different prevalence settings?
- Could you provide a brief check that counterfactual edits are both faithful (influence the decision) and plausible (respect basic clinical constraints), and clarify how RMSE is computed over mixed feature types?
- Could you provide details on which LLM, temperature and seed were used?
- Could you provide more information about the clinician evaluation (number of raters, blinding, inter-rater agreement)?

**Details Of Ethics Concerns:**

This work uses sensitive clinical data for diagnostic-support modeling. Please (i) list IRB/ethics approvals and data-use agreements per cohort; (ii) affirm de-identification and whether any PHI was sent to external services (including the LLM used as a judge - if
cloud-hosted, disclose safeguards or run locally); (iii) clarify sharing plans (code, weights, and any public/synthetic datasets), ensuring no leakage of private data; (iv) discuss automation bias and the risk that counterfactuals may be over-trusted or misinterpreted,
proposing UI/usage guidance and clinician-in-the-loop guardrails; and (v) acknowledge narrow controls (e.g., HCM vs. ATTR) to avoid over-generalization in deployment.

---

### Note · Authors · 2025-12-04

I have read and agree with the venue's withdrawal policy on behalf of myself and my co-authors.